# The Lipidome Fingerprint of Longevity

**DOI:** 10.3390/molecules25184343

**Published:** 2020-09-22

**Authors:** Mariona Jové, Natàlia Mota-Martorell, Irene Pradas, José Daniel Galo-Licona, Meritxell Martín-Gari, Èlia Obis, Joaquim Sol, Reinald Pamplona

**Affiliations:** Department of Experimental Medicine, Lleida University (UdL), Lleida Biomedical Research Institute (IRBLleida), 25198 Lleida, Spain; Mariona.jove@udl.cat (M.J.); nataliamotamartorell@gmail.com (N.M.-M.); irene.pradas@gmail.com (I.P.); jgalolic25@gmail.co (J.D.G.-L.); meritxell.martin@udl.cat (M.M.-G.); eobis@irblleida.cat (È.O.); solcullere@gmail.com (J.S.)

**Keywords:** fatty acids, lipidomics, longevity, membrane unsaturation, peroxidation index

## Abstract

Lipids were determinants in the appearance and evolution of life. Recent studies disclose the existence of a link between lipids and animal longevity. Findings from both comparative studies and genetics and nutritional interventions in invertebrates, vertebrates, and exceptionally long-lived animal species—humans included—demonstrate that both the cell membrane fatty acid profile and lipidome are a species-specific optimized evolutionary adaptation and traits associated with longevity. All these emerging observations point to lipids as a key target to study the molecular mechanisms underlying differences in longevity and suggest the existence of a lipidome profile of long life.

## 1. Introduction

In contrast to life expectancy (also named mean lifespan, and frequently and wrongly termed “longevity”) that may be modified depending on living conditions, maximum longevity (henceforth referred to as “longevity” for practical purposes) is a species-specific feature. For instance, for the world as a whole, human life expectancy has increased more than two-fold in the last century, from 30 years in 1900 to 65 years in 2000 and is estimated to increase to 81 by the end of the 21st century. However, our longevity has remained at approximately 120 years [1]. At the same time, longevity varies widely among animal species with differences of up to 5000 times occurring, for example, between some invertebrates and the mollusk *Arctica islandica*, which at 507 years has the record for animal longevity [2,3], demonstrating the evolutionary plasticity of longevity. Therefore, animal (and human) longevity is flexible, regulated, and evolves rapidly during animal species evolution.

Despite the importance of understanding the mechanisms involved in longevity, the existence and nature of the molecular mechanisms determining and controlling this biological trait remains unclear. Some mechanisms discovered from studies within and among animal species, but not exclusively, include oxidative stress-related pathways [4,5], insulin signaling pathways [6,7,8,9] and mechanistic targets of rapamycin pathway [10,11,12,13]. Notably, genetic, dietary and pharmacological manipulations targeting some of these pathways resulted in more than 10-fold longevity extension in the worm *Caenorhabditis elegans* [14], but only about 1.4-fold longevity extension in rodents [15,16].

Evidence accumulated during the last 25 years demonstrate that lipids are determinant players of animal (and human) longevity. Thus, genomic studies demonstrate that genetic changes in genes controlling lipid metabolism play a role in human longevity [17,18,19,20], as well as in longevity differences among animal species [21,22] and lipid studies on changes in membrane lipid unsaturation and lipidomic profiles (see next sections) among animal species clearly point to lipids as a good potential target for research on molecular mechanisms underlying differences in longevity among animal species.

## 2. Lipids Are Essential for Life

Lipids possess an inherent ability to spontaneously self-organize to generate membranes [23]. This property of lipids was determinant for the origin and early evolution of life [24,25]. The relevance of lipids in life is found in the fact that, without exception, all organisms in the three domains of life (eukaryotes, bacteria, and archaea) have lipid membranes [26]. Indeed, this hydrophobic film of around 30 Å of thickness marks the limit between life and death for cells [27]. The unique trait of the first lipids to form membranes was expanded during evolution to cell signaling as new functional property, and finally to the energy storage [28]. This diversification of functional properties demanded and was supported by an enlargement in the structural and functional diversity of lipid species that conformed early organisms and that evolved toward more complex cell systems. This complexity is reflected in the genetic code, which assigns more than 5% of their genes for the biosynthesis of thousands of different lipids [29], which participate in multitude of pathways and mechanisms that are essential for cell physiology.

In agreement with the current classification system, lipids are categorized into eight groups: sterol (ST) and prenol lipids (PR), saccharolipids (SL) and polyketides (PK), glycerolipids (GL), sphingolipids (SP), glycerophospholipids (GP), and fatty acyls (FA). There are, at present, no consistent assessments on the number of discrete lipid compounds in nature, but the estimations of the cellular lipid profile comprise thousands of different molecular species [30]. This complexity is magnified when the membrane compositional diversity is considered. For instance, the lipid compositional profile varies between animal species, between tissues–organ, between subcellular organelles, between membrane domains, and between the two leaflets of the lipid bilayer [27,31,32,33,34,35,36,37,38,39,40,41,42,43,44,45,46] (Figure 1). However, the complexity increases when this diversity in spatial distribution must be added as a temporal distribution variable. Thus, the lipid profile varies in time according with circadian rhythmicity [47], the physiological state, the vital cycle of an organism, and during evolution [45,48,49,50].

The importance of lipids in cell structure and physiology can also be applied to the pathological condition. Thus, a large number of human pathologies are either etiologically or physiopathologically linked to alterations in lipid homeostasis including genetic disorders [51], and common diseases such as neurodegenerative diseases [52,53], cancer [54,55], non-alcoholic fatty liver [56], cardiovascular disease [57], metabolic risk [58], and obesity and diabetes [53], among several others.

From these findings we can infer that the membrane lipidome (the comprehensive lipid profile) is a structured, complex, dynamic and flexible system, which demands functional plasticity, a location-specific recognition, internal controls to monitor changes and confer stability, and adaptive responses to preserve essential membrane biological properties and cellular functions within physiological limits [37,59]. The result is the formation and maintenance of a specific membrane lipidome. Current evidence from long-lived humans and model organisms demonstrate that the membrane lipidome is associated with longevity, and that it is possible to define a lipid profile of long life.

## 3. Membrane Unsaturation and Longevity

### 3.1. Membrane Unsaturation: A Double-Edged Sword

Membrane unsaturation was an early evolutionary adaptation of biological membranes of the first organisms to maintain the viscosity of the lipid bilayer [60]. Membranes need to stay in the liquid crystalline state to avoid rupture and to provide a homeoviscous environment for the insertion of membrane proteins over the temperature range that a cell can be exposed to [60,61,62]. The affected lipid molecules are essentially fatty acids, the core of most membrane lipids such as glycerophospholipids and sphingolipids, leading to the formation of unsaturated fatty acids. Membrane fatty acids have the dominant influence on the physical and chemical properties of the bilayer they constitute.

Unsaturation of fatty acids carries an inevitable physiological consequence of spontaneous chemical oxidation under aerobic conditions, a process also termed lipid peroxidation [63,64]. It is important to highlight that the degree of sensitivity to oxidation of unsaturated fatty acids is determined by the number of double bonds present in the acyl chain [64,65]. Thus, in saturated or monounsaturated fatty acids, the sensitivity to oxidation is non-existent or practically null, while in unsaturated ones with two or more double bonds their sensitivity to oxidation increases rapidly [66,67]. Thus, from the profile and susceptibility to oxidation of fatty acids of a certain membrane, the so-called peroxidizability or peroxidation index (PI) can be calculated. The peroxidizability index is calculated as PI = [(% monoenoic × 0.025) + (% dienoic × 1) + (% trienoic × 2) + (% tetraenoic × 4) + (% pentaenoic × 6) + (% hexaenoic × 8)]. The higher the PI value, the more susceptible to oxidation is the membrane, and the lower the PI, the greater the resistance of the lipid bilayer to oxidation [66,67].

The lipid peroxidation process occurs in three main phases: an initiation event, chain propagation, and termination [64]. This chemical process is singular so that a single initiation reaction (mediated by e.g., the attack of a free radical) results in 200–400 propagation cycles, rapidly amplifying free radical damage in highly oxidizing unsaturated fatty acid rich environments such as a biological membrane. This radical propagation has been proposed as rate-limiting for the rate of aging, and consequently for longevity, and should be studied more deeply [68]. The termination of the lipid peroxidation process leads to the generation of reactive carbonyl compounds, over a hundred different reactive species have been described, which are mostly cytotoxic, but can also behave as signaling compounds to activate cell antioxidant response systems [37,69,70]. The cytotoxic or regulatory function of these compounds depend on their half-life, reactivity, and abundance. The cytotoxic effects are derived of the ability of these compounds to react chemically and non-enzymatically with functional groups on proteins, nucleic acids, and aminohospholipids, leading to the formation of adducts and cross-links termed advanced lipoxidation endproducts [69,71]. These non-enzymatic modifications induce damaging effects on the molecules, which are modified and have detrimental consequences on the cell physiology [69,71]. In this context, it is plausible to hypothesize that animal species impose clear limitations on the use of unsaturated fatty acids in parallel with increased longevity (see next section).

### 3.2. Membrane Unsaturation and Longevity

The seminal finding of a link between membrane unsaturation and longevity was the work of Pamplona et al. (1996) [72], which revealed that the PI of liver mitochondria from humans, pigeons and rats was negatively correlated with their respective longevities. Later, it was evidenced that this was the case for a wide range of tissues and animal species including both invertebrates and vertebrates (Table 1). All these outcomes obtained at tissue and mitochondrial level were also extended to plasma lipids of mammalian species, including humans [35], demonstrating that the greater the longevity of a species, the lower its plasma PI, reinforcing the idea that lipid unsaturation is a general adaptive trait associated with animal longevity.

While longevity can significantly differ within and between mammals and birds, there can also be important longevity differences within a particular species, as is the case, for instance, within both the wild-derived or the senescence-accelerated mouse (SAM) strains of mice. In both comparisons, the PI of the mouse strain displaying extended longevity was significantly lower that than of the short-lived strain [73,85,103]. Identical results were obtained when membrane unsaturation and PI was evaluated in exceptionally long-lived specimens of mice [95]. Additionally, the comparison between closely-related species with divergent longevity, such as the long-lived white-footed mouse *P. leucopus* and the short-lived *M. musculus,* also displayed a lower PI for the long-lived mouse when compared to the common laboratory mouse [97]. More importantly, two exceptionally long-living mammalian species (naked mole-rats and echidnas) also have membrane fatty acid profiles that are resistant to peroxidative damage as would be predicted from their longevities [32,84,89,104].

Honeybees (*A. mellifera*), flies (*D. melanogaster*), and worms (*C. elegans*) offer new examples of variations in longevity within species that expand previous observations in vertebrates to invertebrates. In honeybees, queens have a longevity that is an order-of-magnitude greater than that of workers [105]. In flies, a comparison is possible between long-living mutants [91] and wild type strains, differing in their longevities [100]. In worms, mutant strains of *C. elegans* that differ 10-fold in their longevities can be compared [92]. The results from all these approaches show that long-lived animals possess a lipid profile of their membranes that is more resistant to oxidation. Thereby, the greater the longevity of the honeybee (the queen), the mutant and strain of fly, and the mutant worm, the lower the membrane unsaturation and more protected are the other cellular components [86,91,92,100], supporting again the idea that membrane composition is an important feature in the determination of longevity.

Different works have shown that bivalves are also excellent models for longevity research [106,107,108] because they are genetically intermediate to conventional invertebrate models of longevity (e.g., flies and worms) and mammalian species, providing a unique window to study the evolution of membrane lipid unsaturation and animal longevity. This taxonomic group both includes the longest-living non-colonial metazoan (the Iceland clam *Arctica islandica*, longevity = 507 years [109]), and the surf clams with animal species of no more than one-year longevity. In this context, the results of a recent work analyzing the membrane fatty acid composition at tissue and mitochondrial level confirmed that long-lived marine bivalves’ mollusks possess peroxidation-resistant membrane lipids [3], analogous to the described findings for mammals, birds, and fishes.

Humans are an exceptionally long-lived species. While the comparison of humans with other animal species reaffirms the relationship between membrane unsaturation and longevity, it should also be interesting to evaluate variations within species by considering centenarians as models of human extreme longevity. If this is the case, the outcomes from two recent studies centered in offspring of long-lived individuals, as well as another specifically focused on centenarians, reinforce the association between membrane lipid unsaturation and human longevity. Thus, the lipid composition of the erythrocytes [90] and plasma [96] of nonagenarian offspring were more resistant than that of matched controls. More interestingly, centenarians possess a plasma lipid composition more resistant to lipid peroxidation than octogenarians and at an adult-like level [48].

Comparative studies have clearly shown that membrane unsaturation is associated with animal longevity. In agreement with this, it was also demonstrated that membrane unsaturation of both total tissue and mitochondrial fractions of long-lived animal species, including long-lived humans, was linked to a lower sensitivity to in vitro and in vivo lipid peroxidation, a lower content of lipid peroxidation-derived products, and a lower concentration of lipoxidation-derived adducts in proteins from tissues such as brain, heart, liver, plasma, and skeletal muscle [5,37,48,66,69]. Reinforcing the idea of attenuated lipoxidation-derived damage in long-lived species, it was also observed that both the accumulation rate of lipofuscin [110], as well as the sensitivity to lipid autoxidation of mammalian tissue homogenates [111], was negatively correlated with the animal longevity.

Therefore, the findings from the comparative approach support a role for membrane unsaturation in the determination of longevity, which seems may be extended to all of the animal kingdom (Figure 2). Reinforcing this concept, studies using exceptionally long-lived animal species do not refute, but instead confirm, the connection between membrane lipid unsaturation and longevity, and support the concept that membrane composition is regulated in a species-specific way. For the particular case of humans, a long-life must be associated with a membrane composition resistant to oxidation.

### 3.3. Longevity Extension by Dietary Interventions are Accompanied by Attenuations of Membrane Unsaturation 

Since correlation does not necessarily mean causation, it is important to know whether cellular components are protected in front of lipid oxidation when membrane unsaturation is modified. With this goal, different experimental studies, based on dietary interventions, were designed to force alterations in membrane lipid composition and to evaluate the changes in both PI and lipoxidative damage of proteins [112,113,114]. The results demonstrate that a low degree of lipid unsaturation of cellular membranes protects the membrane itself and other molecular components against lipid peroxidation-derived products and lipoxidation-derived damage. Importantly, the magnitude of the change in membrane unsaturation is tissue-dependent, and the magnitude of the lipoxidation-derived molecular damage is not proportional to the membrane unsaturation change [112,113,114].

Additional evidence that supports a relationship between membrane unsaturation and longevity comes from dietary interventions that extend longevity in experimental models. Thus, diverse dietary restrictions, such as caloric-, protein-, and methionine restrictions (CR, PR and MetR, respectively), were applied in different animal species (essentially rodents) inducing decreases in the degree of membrane unsaturation, the in vivo and in vitro lipid peroxidation, and the level of lipoxidation products in a diversity of tissues such as liver, heart, kidney and brain [115,116,117,118,119,120,121,122,123,124,125,126,127,128,129,130]. In this line, decreases in the levels of lipofuscin in tissues of *C. elegans* and rodents have also been reported after CR diets [110,131,132,133,134].

From these studies, all of them performed on healthy young experimental animals, it can be also inferred that the magnitude of the change in membrane unsaturation is modest (around 2.5–10%), as would be expected for a highly regulated homeostatic system [59]. In contrast, the effect on lipoxidation-derived protein damage is much higher (about 20–40%), surely due to the additive effect induced by the dietary intervention on mitochondrial free radical generation, which is lowered. Importantly, the changes induced by the dietary manipulations are closely related to the duration and intensity of the applied restriction, with both PR and MetR being more effective than CR.

Additional valuable information was also obtained from long-term interventions throughout the whole animal life designed to chronically modify the lipid profile in order to increase membrane PI and to evaluate its impact in animal longevity. Essentially, two studies can be highlighted, one of them being performed on mice [135], and the other on the worm *C. elegans* [92]. The outcomes from both works demonstrate that the increase in membrane unsaturation is associated with a higher lipid peroxidation, and a significant reduction in animal longevity.

### 3.4. The Potential Mechanistic Basis for the Longevity-Related Differences in Membrane Unsaturation

The molecular basis of the low PI presented by long-lived species is in the redefinition of the lipid profile that conforms membrane composition, with this redefinition being independent of the diet and according to a genotypic pattern [4,5,37,66]. Thereby, there is a general switch affecting the content and nature of unsaturated fatty acids present, from highly unsaturated fatty acids such as docosahexaenoic- (DHA, 22:6n-3), eicosapentaenoic- (EPA, 20:5n-3), and arachidonic (AA, 20:4n-6) acids in short-lived animals to the less unsaturated alpha-linolenic (ALA, 18:3n-3), linoleic (LA, 18:2n-6), and oleic (OA, 18:1n-9) acids in the long-lived ones. Remarkably, the shift produced between fatty acids in the membrane composition is carried out strictly by combining types of unsaturated fatty acids, and not by increasing the content of saturated fatty acids. The result is that the relationship between saturated and unsaturated fatty acids remains stable, the physical-chemical properties of the membrane are not affected, but its susceptibility to oxidation is clearly modified.

From these findings, it can be inferred that the support for the longevity-related differences in fatty acid profiles necessarily implies the unsaturated fatty acid biosynthesis pathways including desaturases, elongases, and peroxisomal beta-oxidation, as well as the diacylation–reacylation cycle. The estimation of elongase and desaturase (delta-5 and delta-6) activities from specific product/substrate ratios indicate that there is a substantial decrease in long-lived species when compared with short-lived ones [4,5,37,66,92]. This can explain why, for instance, LA and ALA decreases and DHA and AA increases from long- to short-lived animals. Therefore, elongation–desaturation pathways are responsible for the availability of unsaturated fatty acids for the de novo synthesis of lipids and later diacylation–reacylation cycle which, in turn, are responsible for the remodeling of glycerophospholipid acyl groups and, consequently, the determination of the lipid profile for a given membrane. In agreement with the proposed mechanism, the study from [21] using a phylogenomic approach certified that, at least in mammals, genes involved in fatty acid biosynthesis (lipoxidation repair, elongases, desaturases, and fatty acid synthase) have collectively undergone, in a specific and significant way, increased selective pressure in long-lived species. These findings were later confirmed [43].

Functional assays, using mutant strains and RNAi to attenuate gene expression, can provide evidence that such genes (for instance, for desaturase or elongases or detoxifying enzymes of reactive compounds derived from lipid peroxidation) are involved in longevity extension. In this context, some studies that observed *C. elegans* deserve a special mention. For instance, it was observed that two long-lived mutant strains of *C. elegans*, daf-2 and age-1, both linked to an attenuation of the insulin-like signaling pathway, showed increased content of monounsaturated fatty acids along with a decreased content in highly unsaturated fatty acids, all leading to the presence of a low PI when compared with the short-lived ones [92]. In another approach, *C. elegans* extended longevity by RNAi suppression of genes encoding either a delta-5 desaturase, fat-4, or two elongases, whereas knockdowns of delta-9 desaturase genes induced a reduction in longevity [92]. In addition, the interference in the expression of genes related to the detoxification of reactive carbonyl species derived from lipid peroxidation (glutathione-S-transferases (GSTs)) induced a significant increase in lipoxidative protein damage concomitantly with a shortening of the worm longevity [136], whereas the overexpression of these enzymes induced a longevity extension [137]. Taken together, these functional findings suggest that reactive carbonyl species derived from lipid peroxidation are causally involved in limiting longevity, and that the modulation of membrane lipid profile to increase resistance to oxidation is one of the mechanisms for longevity extension.

All of these findings have relevance for humans. Humans are a long-lived animal species. Therefore, a trait of its lipidome is the resistance to oxidation. Consequently, human diet should be designed to support and to reinforce this resistance to oxidation, without inducing deficits in essential fatty acids or excess in saturated fatty acids. Surprisingly, the opposite is the case. A trait of the modern human diet is the excess in PUFA consumption, with an additional imbalance between n3 and n6 PUFA due to the increased content of the latter, which favor pro-inflammatory conditions that are the basis of diseases such as insulin resistance, obesity, cancer and neurodegenerative diseases [28]. Humans possess the metabolic machinery to synthesize PUFA when needed. Thus, the key is to offer balanced n3 and n6 substrates (the essential fatty acids, which have low unsaturation degree and are resistant to oxidation) instead of the products (PUFA) that are highly susceptible to oxidation. It is imperative to improve people’s diet and to define both the optimal fatty acid amount and type to enhance the membrane fatty acid profile for a long life.

### 3.5. The Lipid Bilayer as Dynamic Structural Adaptive System

Long-lived animal species (and humans) possess a low degree of membrane unsaturation based on the redefinition of the unsaturated fatty acid profile. This elegant evolutionary strategy allows decrease in the sensitivity to lipid peroxidation without altering fluidity, a basic property of cellular membranes. This occurs because membrane fluidity significantly increases with the incorporation of the first (usually around the center of the fatty acid) and less with the second double bond, whereas additional double bonds (usually toward the extreme of the acyl chain) cause few further variations in fluidity [138]. In contrast, peroxidation index increases with the number of double bonds, irrespective of its location [65]. Thus, the switch in the fatty acid profile from short- to long-lived animal achieves a lesser sensitivity to lipid peroxidation at the same time as maintaining the membrane fluidity. This idea, evocative of membrane acclimation to different environments at unsaturated fatty acid level in bacteria and poikilotherm animals, was defined as the homeoviscous longevity adaptation theory [66] and confirms the biological membranes as a dynamic structural adaptive system.

## 4. Lipidomics of Longevity

Fatty acids are essential components of the structural diversity of lipids of all cell membranes. Indeed, by permuting headgroups and fatty acids, more than 10,000 lipid molecular species can be built [139]. Recent developments in mass spectrometry- (MS)-based lipidomics offer the means to unambiguously and accurately detect, identify and quantify a huge number of lipid molecular species, as well as to profile large-scale changes in lipid composition. Current data also suggest that the lipidomic profile is an optimized trait associated with longevity.

The comprehensive lipidomic profiles and lipid concentrations identify the animal species, i.e., the lipidome is species-specific (see Figure 1), as well as the long-lived animal species [35,41,43,140]. These comparative studies were carried out in a diversity of tissues such as cerebellum, cortex, heart, kidney, liver, muscle, and plasma, and exclusively in mammalian species including exceptionally long-lived animal species such as the naked mole-rat, bats and humans. Lipidomics studies also define exceptionally long-lived humans (centenarians) [48,96,141,142].

The findings from these studies reveal the presence of common lipidomic features, which accurately predict animal longevity, and the centenarian condition. More specifically, the relationship between lipids and longevity were ascribed to lower or higher lipid concentrations in species of specific lipid categories such as glycerolipids (diacylglycerols, DAGs), glycerophospholipids (GPs) (particularly ether lipids), sphingolipids (SPs), and long chain free fatty acids (LC-FFAs) [35,43,140]. Although the functional significance of most of these changes remains to be elucidated, some considerations can be exposed.

DAGs are precursors for the de novo GP biosynthesis, one of the main components of cell membranes. A recent study [140] revealed that DAGs negatively correlate with animal longevity, suggesting that long-lived species (including humans) possess a lower rate of GP biosynthesis through the de novo pathway, likely due to a lower rate of membrane lipid exchange.

Ether lipids are a subclass of GPs, mostly present as phosphatidylcholine (PC) and phosphatidylethanolamine (PE) species [143]. Ether lipids have an alkyl chain attached by an ether bond at the sn-1 position of the glycerol backbone. The sn-2 position of ether lipids has an ester-linked acyl chain, as in diacyl phospholipids. Some ether-linked phospholipids, called alkenyl-acylphospholipids, contain a cis or Z double bond adjacent to the ether linkage and are commonly referred to as plasmalogens [143]. Ether lipids represent about 20% of the total GP pool in mammals and have a tissue-dependent distribution [144]. Findings from studies in animal models of exceptional longevity such as *A. islandica* [3] and the naked mole-rat [32], as well as *C. elegans* [145], suggest an association between plasmalogen content and animal longevity. For instance, the naked mole-rat had no DHA-containing plasmalogens but had much higher levels of total PC plasmalogens. For bivalves, no correlation between plasmalogens content and longevity comparing five different bivalves’ species was found. For worms, *C. elegans* strains carrying loss-of-function mutations in genes encoding protein required for ether lipid biosynthesis demonstrated a shorter lifespan, and a decreased resistance to oxidative stress. In humans, lipidomic studies describe different plasma concentration of ether lipid species in centenarian [48,142] and in middle-aged offspring of nonagenarians [96], suggesting an ether lipid signature in long-lived humans. This signature is featured by higher level of alkyl forms derived from PC and decreased content in alkenyl forms from PE [142]. Remarkably, the compositional pattern in fatty acids of these ether lipids is specific, resulting in an ether lipid profile in centenarians that is more resistant to lipid peroxidation [142]. The physiological role of ether lipids, and specially plasmalogens, is associated with their function as membrane components [143]. However, interestingly, an antioxidant role has also been attributed to plasmalogens, which, similar to a scavenger, protects membrane unsaturation against oxidation [146]. Therefore, it was proposed that the ether lipid signature is a specific trait of long-lived humans.

Another important category of lipids highly conserved in all eukaryotic cells is sphingolipids (SPs). SPs possess a natural predisposition to the formation of lipid domains [147]. In the metabolism of sphingolipid, ceramide plays the role of a central metabolic hub [148] from which a wide diversity of structural and bioactive SP species is synthesized. Sphingolipids regulate a number of functions at membrane and cellular level. Its relationship with longevity proceeds from studies in model organisms such as yeast and flies. In these animal species, deletions or mutations of genes affecting ceramide biosynthesis or metabolism were able to significantly modify longevity [149,150,151]. Lipidomic studies in animal models, such as *S. cerevisiae* [152], *C. elegans* [153], the naked mole-rat [154], and mammals [35,43], also suggest an association between sphingolipid content and animal longevity. Specifically, a low sphingolipid content, particularly for sulfatides, ceramides and glycosphingolipids, seems to be a feature of long-lived animal species. Lipidomic studies in long-lived humans are limited and results are not conclusive [48,96,141,155,156,157].

Finally, the importance of lipid metabolism in determining longevity was also verified with respect to the plasma concentrations of LC-FFAs in different mammals, including humans [35]. The results reveal that long-lived species present lower plasma LC-FFA concentration, PI, and lipid peroxidation-derived products content, conferring to plasma, analogous to cell membranes, superior resistance to oxidation. Additionally, LC-FFAs are also essential by covering mammalian energy needs [158]. Interestingly, adipose tissue is the main source for the liberation of FFA into plasma. Consequently, the obtained results from the comparative approach suggest an evolutionary adaptation for adipose tissue regulating the kind of fatty acids stored and released to plasma. The result would be an adipose tissue with a composition and activity that is species-specific and designed to maintain a lower degree of unsaturation in long-lived species. Another important consequence derives from the fact that LC-FFAs also have signaling properties in several physiological processes [159,160,161]. Among them, a particularly relevant property is their direct effect on insulin secretion. Thus, it is plausible to hypothesize that long-lived mammals and human exhibit a lower LC-FFA concentration to maintain an attenuated activity of the insulin signaling pathway, which is in agreement with the consideration of insulin signaling pathway as an evolutionary conserved mechanism involved in the determination of animal longevity [162,163]. In this context, the lower concentration of LC-FFAs observed in humans could be considered an evolutionary adaptive response that could explain pathological processes like insulin resistance, which is linked to an increased LC-FFA plasma concentration [158] and, in turn, to a shortened longevity.

## 5. Conclusions

The emergence of lipids in the biosphere was key for the origin and evolution of life because of their capacity to generate membranes. Membrane composition contributed and participated in the acquisition of different longevities of animal species, which was translated in the generation of membranes with a particular lipid composition in long-lived animal species. In particular, membranes from longevous animal species and humans are featured by a low degree of unsaturation due to a shift in the fatty acid profile from highly unsaturated to fatty acids with a low number of double bonds leading to membranes resistant to oxidation, as well as a lipidomic signature species-specific affecting particular lipid molecular species. Membrane unsaturation and lipidomic signatures of longevity are conserved mechanisms during evolution for longevity determination. Therefore, there is a lipidome profile of long life.

## Figures and Tables

**Figure 1 molecules-25-04343-f001:**
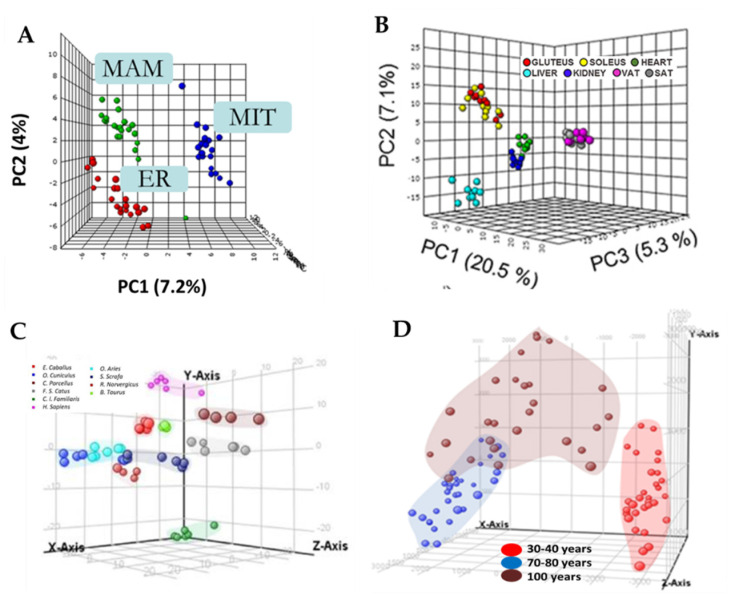
Specific lipidomic profiles at subcellular, tissue, and animal species level. (**A**) Specific lipidomic profile defines different subcellular components such as endoplasmic reticulum (ER), mitochondrial-associated membranes (MAM), and mitochondrion (MIT). (**B**) Specific lipidomic profile for rat tissues. Principal component analysis (PCA) representation of the lipidome of all the tissues in positive ionization. Modified with permission from [46]. (**C**) Specific plasma lipidomic profiles for mammalian species. PCA representation (positive ionization molecules) showing that plasma lipidomic profiles are species-specific. Modified with permission from [35]. (**D**) Human extreme longevity as model of healthy aging. PCA (positive ionization) revealed differences in adults, aged, and centenarian plasma lipidomic profiles. Modified with permission from [48].

**Figure 2 molecules-25-04343-f002:**
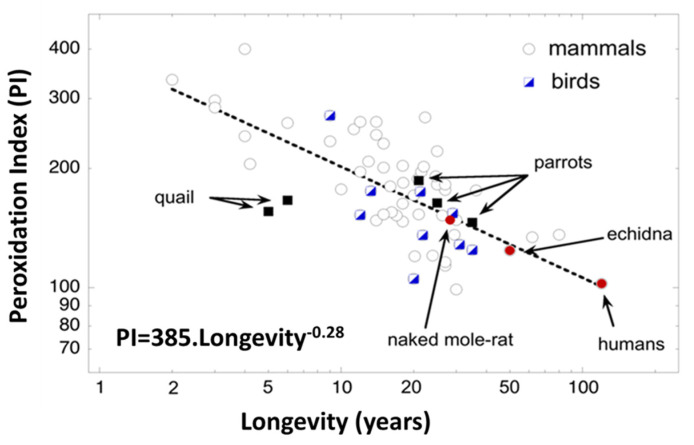
Relationship between peroxidation index (PI) and longevity of skeletal muscle phospholipids for mammalians and birds, pointing out exceptionally long-lived animal species. Modified with permission from [28].

**Table 1 molecules-25-04343-t001:** Updated list of comparative studies on the relationship between membrane peroxidation index (PI) and longevity in invertebrates and vertebrates, including humans.

Animal Species	Longevity	PI * Values	Tissue (or Subcellular Organelle)	PI in Long-Lived Species	Ref
Human, pigeon, rat	120, 35, 4 yrs	45, 67, 97	Liver mitochondria	Lower	[72]
SAM-R/1 vs. SAM-P/1 mice	1.8, 1.2 yrs	n.a. (1)	Liver	Lower	[73]
8 mammals	From 46 to 3.5 yrs	From 71 to 145	Liver mitochondria	Lower	[74]
Pigeon vs. rat	35, 4 yrs	80, 145	Heart mitochondria	Lower	[75]
Canary vs. mouse	24, 3.5 yrs	180, 230 (2)	Heart	Lower	[76]
Parakeet vs. mouse	21, 3.5 yrs	150, 230 (2)	Heart	Lower	[76]
Pigeon vs. rat	35, 4 yrs	119, 153	Liver mitochondria	Lower	[77]
Pigeon vs. rat	35, 4 yrs	95, 151	Liver microsomes	Lower	[77]
Pigeon vs. rat	35, 4 yrs	131, 122	Heart mitochondria	Unchanged	[77]
Pigeon vs. rat	35, 4 yrs	110, 126	Heart microsomes	Lower	[77]
8 mammals	From 46 to 3.5 yrs	From 70 to 246	Heart	Lower	[78]
7 mammals	From 46 to 3.5 yrs	From 65 to 158	Liver	Lower	[79]
8 mammals	From 46 to 3.5 yrs	For PC: from 122 to 152 (2). For PE: from 140 to 220 (2). For CL: from 115 to 132 (2).	Liver mitochondria (PC, PE, and CL fractions)	Lower	[80]
Pigeon vs. rat	35, 4 yrs	60, 120	Skeletal muscle	Lower	[81]
Canary, parakeet, mouse	24, 21, 3.5 yrs	225, 223, 210	Whole brain	Unchanged	[82]
8 mammals	From 46 to 3.5 yrs	From 70 to 247	Heart	Lower	[83]
Naked-mole rat vs. mouse	32, 3.5 yrs	150, 225	Skeletal muscle	Lower	[84]
Naked-mole rat vs. mouse	32, 3.5 yrs	160, 265	Heart	Lower	[84]
Naked-mole rat vs. mouse	32, 3.5 yrs	130, 225	Kidney	Lower	[84]
Naked-mole rat vs. mouse	32, 3.5 yrs	160, 170	Whole Brain	Unchanged	[84]
Naked-mole rat vs. mouse	32, 3.5 yrs	130, 185	Liver	Lower	[84]
Naked-mole rat vs. mouse	32, 3.5 yrs	135, 180	Liver mitochondria	Lower	[84]
Idaho, Majuro and wild-type mice	3.9, 3.5, 3.3 yrs	172, 178, 190	Liver	Lower	[85]
Idaho, Majuro and wild-type mice	3.9, 3.5, 3.3 yrs	213, 263, 274	Skeletal muscle	Lower	[85]
12 mammals and 9 birds	From 120 to 3.5 yrs	From 130 to 350	Skeletal muscle	Lower	[67]
10 mammals and 8 birds	From 120 to 3.5 yrs	From 45 to 250	Liver mitochondria	Lower	[67]
Queen honey bees vs. workers	>5 yrs, 75–135 days	15, 45	Head, thorax, abdomen	Lower	[86]
42 mammals	From 70 to 2 yrs	From 10 to 42 (4)	Skeletal muscle	Lower (3)	[87]
13 bird species (9 petrels vs. 4 fowl)	41, 9 yrs	100, 160	Heart	Lower	[88]
Echidna vs. mammals (5)	54, 4 yrs	121, 350	Skeletal muscle	Lower	[89]
Echidna vs. mammals (5)	54, 4 yrs	100, 160	Liver	Lower	[89]
Echidna vs. mammals (5)	54, 4 yrs	79, 150	Liver mitochondria	Lower	[89]
Humans (nonagenarian offspring vs. matched control)	67, 68 yrs	63, 84	Erythrocytes	Lower	[90]
*D. melanogaster* (long-lived mutant strains)	87, 71 days	9, 13	Whole fly and mitochondria	Lower	[91]
*C. elegans* (long-lived mutant strains)	From 170 to 18 days	From 81 to 140	Whole worm	Lower	[92]
Pigeon vs. rat	35, 4 yrs	160, 210	Heart	Lower	[93]
Pigeon vs. rat	35, 4 yrs	130, 160	Heart mitochondria	Lower	[93]
Pigeon vs. rat	35, 4 yrs	150, 200	Pectoral muscle	Lower	[93]
Pigeon vs. rat	35, 4 yrs	130, 200	Pect. Muscle mitochondria	Lower	[93]
Pigeon vs. rat	35, 4 yrs	120, 180	Liver	Lower	[93]
Pigeon vs. rat	35, 4 yrs	110, 180	Liver mitochondria	Lower	[93]
Pigeon vs. rat	35, 4 yrs	140, 220	Leg muscle	Lower	[93]
Pigeon vs. rat	35, 4 yrs	120, 170	Kidney	Lower	[93]
Pigeon vs. rat	35, 4 yrs	80, 150	Erythrocytes	Lower	[93]
Pigeon vs. rat	35, 4 yrs	190, 170	Whole Brain	Unchanged	[93]
5 bivalve species	From 507 to 28 yrs	From 175 to 268	Gill mitochondria	Lower	[3]
5 bivalve species	From 507 to 28 yrs	From 172 to 222	Gill cell debris	Lower	[3]
5 birds (3 long-lived parrots vs. 2 short-lived quails)	27, 5.5 yrs	150, 150 (6)	Heart	Unchanged	[94]
5 birds (3 long-lived parrots vs. 2 short-lived quails)	27, 5.5 yrs	160, 160 (6)	Leg muscle	Unchanged	[94]
5 birds (3 long-lived parrots vs. 2 short-lived quails)	27, 5.5 yrs	140, 140 (6)	Kidney	Unchanged	[94]
5 birds (3 long-lived parrots vs. 2 short-lived quails)	27, 5.5 yrs	190, 190 (6)	brain	Unchanged	[94]
5 birds (3 long-lived parrots vs. 2 short-lived quails)	27, 5.5 yrs	100, 100 (6)	Erythrocytes	Unchanged	[94]
5 birds (3 long-lived parrots vs. 2 short-lived quails)	27, 5.5 yrs	160, 160 (6)	Pectoral muscle	Unchanged	[94]
5 birds (3 long-lived parrots vs. 2 short-lived quails)	27, 5.5 yrs	130, 130 (6)	Liver	Unchanged	[94]
Exceptionally old, old, and adult mice	128, 76, 28 weeks	171, 199, 190 (for brain) 181, 197, 169 (for spleen)	Brain, spleen	Lower	[95]
Exceptionally old, old, and adult mice	128, 76, 28 weeks	171, 199, 190 (for brain) 181, 197, 169 (for spleen)	Brain, spleen	Lower	[95]
Humans (nonagenarian offspring vs. matched control)	59, 58 yrs	n.a.	Plasma	Lower (7)	[96]
Long-lived vs. short-live mouse (*P. leucopus* vs. *M. musculus*)	8, 3.5 yrs	n.a.	Skeletal muscle mitochondria	Lower (8)	[97]
Long-lived Ames dwarf mice vs. normal-sized littermates	4.9, 3.5 yrs	235, 284	Skeletal muscle	Lower	[98]
Long-lived Ames dwarf mice vs. normal-sized littermates	4.9, 3.5 yrs	266, 331	Heart	Lower	[98]
Long-lived Ames dwarf mice vs. normal-sized littermates	4.9, 3.5 yrs	175, 210	Liver	Lower	[98]
Long-lived Ames dwarf mice vs. normal-sized littermates	4.9, 3.5 yrs	185, 243	Liver mitochondria	Lower	[98]
Long-lived Ames dwarf mice vs. normal-sized littermates	4.9, 3.5 yrs	180, 171	Whole Brain	Unchanged	[98]
11 mammals, including humans	From 120 to 3.5 yrs	n.a.	Plasma	Lower (9)	[35]
107 bird species	From 45 to 5 yrs	From 80 to 50	Liver	Higher (10)	[99]
Human, pig, mouse	120, 27, 3.5 yrs	70, 80, 225	Skeletal muscle	Lower (11)	[41]
Human, pig, mouse	120, 27, 3.5 yrs	100, 125, 130	Liver	Lower (11)	[41]
Human, pig, mouse	120, 27, 3.5 yrs	130, 160, 165	Brain	Lower (11)	[41]
Centenarians, octogenarians, and adults	100, 75, 30 yrs	64, 72, 66	Plasma	Lower	[48]
35 mammals (primates, rodents and bats)	From 120 to 4 yrs	n.a.	6 tissues (liver, muscle, kidney, heart, cortex, cerebellum)	Lower (12)	[43]
*D. melanogaster* strains (Long-lived Oregon R vs. short-lived Dahomey flies)	74, 49 days	10, 17	Whole fly	Lower	[100]
Honey bees (long-lived queens vs. short-lived worker bees)	>5 yrs, 75–135 days	12, 27	Whole honey bee	Lower	[101]
4 fish species (long-lived *Aphyosemion australe* vs. short-lived *Nothobranchius korthausae*, *N. rachobii* and *N. guentheri*)	3 yrs, and 80, 63, and 53 weeks	400, 419, 437, 429 (for PE); 268, 286, 289, 306 (for PS)	Whole fish	Lower	[102]

* The peroxidizability index was calculated as PI = [(% monoenoic × 0.025) + (% dienoic × 1) + (% trienoic × 2) + (% tetraenoic × 4) + (% pentaenoic × 6) + (% hexaenoic × 8)]. Abbreviations: n.a., not available; PC, phosphatidylcholine; PE, phosphatidylethanolamine; PS, phosphatidylserine; CL, cardiolipin. (1) Data available as “lipid peroxidation”. (2) Data available as double bond index (DBI). The density of double bonds in the membrane is calculated with the double bond index, DBI = [(1 × Σmol% monoenoic) + (2 × Σmol% dienoic) + (3 × Σmol% trienoic) + (4 × Σmol% tetraenoic) + (5 × Σmol% pentaenoic) + (6 × Σmol% hexaenoic)]. (3) Results obtained after correction for body weight and phylogeny showed that longevity decreases as the ratio of n-3 to n-6 PUFAs increases. No relation between longevity and PI was found. (4) Interspecies range for PUFA n3 content (%) is indicated. (5) Only PI value for rat is indicated. (6) Average values from 3 parrot species (long-lived) and from 2 quail species (short lived). (7) Based on a MUFA/PUFA ratio significantly higher in offspring group. (8) From isoprotane concentration (as measurement of lipid peroxidation). (9) From plasma concentration of lipid peroxidation products. (10) Long-lived bird species are associated with a lower both PUFA and PUFAn6 content, and a higher MUFA content, DBI and PI. (11) PI is calculated from fatty acid composition of 3 phospholipid fractions (phosphatidylcholine+phosphatidylethanolamine+phosphatidylserine). (12) Based on the average of the number of double bonds for specific lipids categories.

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
