# Peer review of "The Lipidome Fingerprint of Longevity"

_molecules, 2020, doi:10.3390/molecules25184343_

Round 1
Reviewer 1 Report
In this review manuscript, the authors elaborate a theory by comparing longevity with the tissue and fluid content in unsaturated lipids and establishing a negative association between these two parameters. The association is supported by a rich list of studies, mostly performed inter-species but also by comparing intra-specific groups. They attribute this association to the oxidizable character of double bonds, and the deleterious effects of lipid peroxidation on cell function.
Some minor comments:
- The peroxidation index (PI) is presented on page 3 and appears throughout the text as the main parameter in comparisons. It would be good to remind how this PI is calculated, and this should be explicitly shown around line 111, when PI is defined.
- Table 1, would be more informative if two additional columns were included, indicating the PI and the longevity values, respectively, corresponding to the groups compared in each case.
- On line 226, please specify what kind of “protein damage” is considered.
- The authors could comment on the pretended benefits of currently praised highly unsaturated diets in the context of their statement.
- In the paragraph beginning on line 318, a clear definition of “plasmalogen” is needed.
Author Response
Responses to reviewers
R/ We would like to thank reviewers for their interest, helpful comments and suggestions. We have corrected the manuscript (MS) in order to address them, which we feel has clarified and improved it. New text parts (or changes) are highlighted throughout the MS in yellow. The detailed answers to the specific points are given below.
Reviewer 1
In this review manuscript, the authors elaborate a theory by comparing longevity with the tissue and fluid content in unsaturated lipids and establishing a negative association between these two parameters. The association is supported by a rich list of studies, mostly performed inter-species but also by comparing intra-specific groups. They attribute this association to the oxidizable character of double bonds, and the deleterious effects of lipid peroxidation on cell function.
Some minor comments:
The peroxidation index (PI) is presented on page 3 and appears throughout the text as the main parameter in comparisons. It would be good to remind how this PI is calculated, and this should be explicitly shown around line 111, when PI is defined.
R/ In accordance with the reviewer comment, how PI is calculated has been incorporated in the new manuscript version (see lines 125-127, highlighted in yellow).
Table 1, would be more informative if two additional columns were included, indicating the PI and the longevity values, respectively, corresponding to the groups compared in each case.
R/ In accordance with the reviewer comment, Table 1 has been modified including two additional columns containing information about longevities of animal species studied, as well as their PI values (see new manuscript version). Furthermore, additional information was introduced to improve table.
On line 226, please specify what kind of “protein damage” is considered.
R/ In accordance with the reviewer comment, it is clarified that protein damage in referred to lipoxidation-derived protein damage. See lines 259-260, 271-272, and 323 of the new manuscript version (highlighted in yellow).
The authors could comment on the pretended benefits of currently praised highly unsaturated diets in the context of their statement.
R/ In accordance with the reviewer comment, a new paragraph has been added to discuss briefly the relevance of described findings for humans. See lines 328-336 of the new manuscript version (highlighted in yellow).
In the paragraph beginning on line 318, a clear definition of “plasmalogen” is needed.
R/ In accordance with reviewer comment, a brief definition of plasmalogen has been incorporated in the new manuscript version (see line 383-387, highlighted in yellow).
Those authors did a lot of groundwork before this review, and the authors did a good job to summarize the relationship between membrane unsaturation and longevity.
Reviewer 2 Report
Those authors did a lot of groundwork before this review, and the authors did a good job to summarize the relationship between membrane unsaturation and longevity.
I have two suggestions:
1. I think the title "Lipids for Long Life " is too big. To be frank, it misled me before I read the article. If you want to use this title, you should pay more attention to lipid metabolism, lipid signaling molecules, and so on. I suggest changing the title to "Membrane unsaturation for Long Life"
2. I think the three-dimensional scatter plots (Figure 1) are not concise and precise enough. For example, I cannot distinguish the position of the MAN,MIT, and ER position on component 2 axis in "Top left" of Fig 1. I read the origin articles for Figure 1 (ref 35,48, 46) too, they showed the same problem. I guess it is a software problem, but I highly suggested you add some vertical lines for each point to eliminate misunderstandings. You can use the R language to draw three-dimensional scatter plots with vertical lines. (The detailed methods can be easily found on the book "R Graphics Cookbook"). Otherwise, it is really hard for me to read the figure correctly.
In summary, I love this review and the authors are experts in this field. Minor revision is needed before publication.
Author Response
Responses to reviewers
R/ We would like to thank reviewers for their interest, helpful comments and suggestions. We have corrected the manuscript (MS) in order to address them, which we feel has clarified and improved it. New text parts (or changes) are highlighted throughout the MS in yellow. The detailed answers to the specific points are given below.
Reviewer 2
I have two suggestions:
1.I think the title "Lipids for Long Life " is too big. To be frank, it misled me before I read the article. If you want to use this title, you should pay more attention to lipid metabolism, lipid signaling molecules, and so on. I suggest changing the title to "Membrane unsaturation for Long Life".
R/ We thank the reviewer's suggestion. We share with the reviewer that perhaps the title can lead to confusion of what to expect from the review. Therefore, we agree to change the title. About this, to comment you that although the idea of the membrane unsaturation represents a substantial part of the manuscript, we think that it is also important the section dedicated to lipid species and longevity. For this reason, we ask the reviewer to allow us to modify title to a new version: ‘The lipidome fingerprint of longevity’.
2.I think the three-dimensional scatter plots (Figure 1) are not concise and precise enough. For example, I cannot distinguish the position of the MAN, MIT, and ER position on component 2 axis in "Top left" of Fig 1. I read the origin articles for Figure 1 (ref 35,48, 46) too, they showed the same problem. I guess it is a software problem, but I highly suggested you add some vertical lines for each point to eliminate misunderstandings. You can use the R language to draw three-dimensional scatter plots with vertical lines. (The detailed methods can be easily found on the book "R Graphics Cookbook"). Otherwise, it is really hard for me to read the figure correctly.
R/We thank the reviewer for the suggestion. We searched for the R book he/she has recommended and we understand that he/she would like us to represent a Figure where the specific scores are easier to identify as following:
See attached file
However, the main goal of Figure 1 is to visualize that those samples from the same class group together rather than to know the exact position of each sample, which would also require to describe the loadings of each lipid for each component. There is specific metabolomics freeware and software as Metabolanalyst (Pang, Z., Chong, J., Li, S. and Xia, J. (2020) MetaboAnalystR 3.0: Toward an Optimized Workflow for Global Metabolomics. Metabolites 10(5) 186) and MassHunter Mass Profiler (Agilent Technologies) which have the option of performing a general PCA to evaluate the global metabolome or lipidome of a specific sample giving us the information about the clusterization of the samples of the different groups. The four figures represented in Figure1 are performed using this kind of metabolomics software with the only objective of represent the variability of each sample and visualize the grouping of the samples analyzed. For this reason, we ask the reviewer to allow us to maintain the current style of Figure 1. Reviewing Figure 1, we noticed that the axes in the first image were not clear enough so we have enhanced this point.
In summary, I love this review and the authors are experts in this field. Minor revision is needed before publication.